# Nanosecond pulsed electric fields increase antibiotic susceptibility in methicillin-resistant *Staphylococcus aureus*

Alexandra E. Chittams-Miles,[1,2] Areej Malik,[2,3] Erin B. Purcell,[3] Claudia Muratori[1,4]

**ABSTRACT** *Staphylococcus aureus* is the leading cause of skin and soft-tissue infections (SSTIs). SSTIs caused by bacteria resistant to antimicrobials, such as methicillin-resistant *S. aureus* (MRSA), are increasing in incidence and have led to higher rates of hospitalization. In this study, we measured MRSA inactivation by nanosecond pulsed electric fields (nsPEF), a promising new cell ablation technology. Our results show that treatment with 120 pulses of 600 ns duration (28 kV/cm, 1 Hz), caused modest inactivation, indicating cellular damage. We anticipated that the perturbation created by nsPEF could increase antibiotic efficacy if nsPEF were applied as a co-treatment. To test this hypothesis, we used three antibiotics approved to treat SSTI, daptomycin, doxycycline, and vancomycin, and compared the cytotoxic effects of these antibiotics administered either before or after nsPEF. Co-treatment with nsPEF and daptomycin greatly potentiated the effects of each monotherapy regardless of their order. Conversely, the sensitivity of MRSA to both doxycycline and vancomycin was increased only when nsPEF preceded the antibiotic incubation. Finally, MRSA cells grown in biofilms were efficiently killed by co-treatment with nsPEF/vancomycin, suggesting that their mutual enhancement is maintained even when treating sessile communities known for their inherent antimicrobial resistance. Altogether our results show that MRSA perturbation by nsPEF potentiates the effect of multiple antibiotics and that the order of the combined treatment can have a major impact on efficacy. Since SSTIs are accessible for physical interventions such as nsPEF stimulus, combinatorial treatments could be used to increase the efficacy of antibiotics used to treat such infections.

**IMPORTANCE** We have found that treatment with short electric pulses potentiates the effects of multiple antibiotics against methicillin-resistant *Staphylococcus aureus*. By reducing the dose of antibiotic necessary to be effective, co-treatment with electric pulses could amplify the effects of standard antibiotic dosing to treat *S. aureus* infections such as skin and soft-tissue infections (SSTIs). SSTIs are accessible to physical intervention and are good candidates for electric pulse co-treatment, which could be adopted as a step-in wound and abscess debridement.

**KEYWORDS** electroporation, skin infection, MRSA, *Staphylococcus aureus*

Address correspondence to Erin B. Purcell, epurcell@odu.edu, or Claudia Muratori, cmurator@odu.edu.

Alexandra E. Chittams-Miles and Areej Malik contributed equally to this article. Author order was determined by alphabetical order.

The authors declare no conflict of interest.

See the funding table on p. 14.

The Gram-positive opportunistic pathogen *Staphylococcus aureus* (*S. aureus*) is the leading cause of skin and soft-tissue infections (SSTIs) in the United States (1, 2). Patients with ulcers, commonly resulting from advanced complications of injuries, recent surgery, or indwelling medical devices, are particularly at risk. SSTIs range from superficial infections such as impetigo, cellulitis, simple abscesses, and furuncles to deeper and more severe infections such as necrotizing infections, infected ulcers, infected burns, and major abscesses. Moreover, diabetic foot infections are similar to SSTIs in pathophysiology, microbiology, and treatment and can be seen as a subset of SSTIs (3). SSTIs are common in ambulatory and inpatient settings, accounting for more

than 14 million outpatient visits and 850,000 hospitalizations annually in the United States alone (4). Between 7% and 10% of hospitalized patients have SSTIs, which are often hospital-acquired infections that complicate treatment of the original ailment (5, 6).

Treatment of SSTIs varies based on clinical severity, patient comorbidities, admission status, and diagnosis. Uncomplicated SSTIs are treated with topical or oral antibiotics, while severe SSTIs require early aggressive surgical debridement accompanied by antibiotic interventions (7). The history of *S. aureus* treatment is marked by the development of resistance to each new class of antimicrobial drugs, including penicillin, sulfonamides, tetracyclines, glycopeptides, and others, complicating therapy (8). Methicillin, which inhibits bacterial cell wall synthesis, was once a front-line treatment for *S. aureus* infections but this resulted in the development of methicillin-resistant strains of *S. aureus* (MRSA) (9). First reported in the 1960 s (10), MRSA has become increasingly prevalent since the 1980 s (11, 12) and is now endemic in many hospitals and even epidemic in some, with methicillin resistance present in approximately 30% of all *S. aureus* infections in the United States (12). Vancomycin is the only antibiotic that can consistently successfully treat MRSA (13). However, the emergence of *S. aureus* infection with intermediate resistance to vancomycin in the United States suggests that *S. aureus* strains are constantly evolving, and full resistance may develop (14). Novel approaches to tackle this problem are urgently required.

Recently, the potential use of physical methods as an aid to antibiotics in the battle against bacterial pathogens has received greater attention: photodynamic therapy (15, 16), ultrasound therapy (17–19), thermotherapy (20), and weak electric currents (21–25) are all being tested as treatment modalities against pathogenic microorganisms. The major drawback of these methods is their low therapeutic index due to high levels of heating (26) or production of reactive oxygen species, both of which can damage the tissues in and around the target area (27). Moreover, these methods usually require a prolonged exposure time and, for photodynamic therapy, a photosensitizer. As of today, none of the above-mentioned methods has matured into an approved treatment modality against bacterial pathogens.

Pulsed electric fields (PEF) are successfully used in a wide range of clinical applications from cancer therapy to cardiac ablation (28, 29). The application of PEF disrupts cell plasma membranes in mammalian and bacterial cells and has been used for decades to promote bacterial uptake of exogenous DNA in laboratory settings (30, 31). This disruption of membrane barrier function, called electroporation, leads to multiple cytophysiological effects, including calcium ($Ca^{2+}$) overload, efflux of ATP and other metabolites, and disturbances in transmembrane ion gradients ($Na^+$, $K^+$, and $Cl^-$) required for the maintenance of membrane resting potential and for osmotic and cell volume regulation (32–40). The biological effects of PEF can be tuned by adjusting pulse parameters such as pulse number, duration, and amplitude. In most protocols, the rate of energy deposition is controlled so that the concurrent Joule heating, the process where the energy of an electric current is converted into heat as it flows through a resistance, does not cause thermal damage (36, 41–44). When the electroporative damage exceeds the cell repair capacity, PEF treatments cause cell death. The ability of PEF to inactivate microorganisms has been known for over 60 years (45). Indeed, PEF are among the most promising microbial inactivation methods for liquid food (46), and wastewater (47, 48). Moreover, due to the physical nature of the main underlying mechanism—formation of aqueous pores in the plasma membrane—bacteria cannot easily develop genetically encoded resistance against it.

Recent research has extended PEF treatments to the nanosecond duration range (nsPEF). Because nsPEF use much shorter pulses (as low as 10 ns), higher voltages can be applied with minimal thermal effects. Compared to micro- and millisecond pulses, permeabilization by nsPEF does not rely on charge movement or capacitive charging, resulting in a much more uniform/less localized poration pattern, or so-called "supra-electroporation" (49, 50). Moreover, nanosecond pulses permeabilize not only the outer

membrane of the cell but also the intracellular membranes of eukaryotic organelles, such as the endoplasmic reticulum and the mitochondria (38, 39, 51–54). nsPEF create nanopores with cross-sections less than 1.5 nm with resulting cell permeabilization lifetimes on the order of seconds or minutes (37, 55, 56). These nanopores allow efflux of cellular contents but can also increase the uptake of exogenous substances such as drugs (57–59). Finally, the treatment of human tissues with nsPEF minimizes neuromuscular stimulation. Pulses in the micro- to millisecond duration range trigger neuromuscular excitation, which causes severe pain and involuntary muscle contractions (60–65). *In silico* models show that standard 100 µs electric pulses excite nerves at an electric field strength ~1,000-fold lower than that needed for ablation (66). While anesthesia and muscle relaxants offer a partial solution, the optimal solution is to minimize nerve excitation in the first place. Both theoretical and experimental research demonstrate that shortening the pulse duration into the nanosecond range decreases the neuromuscular response to PEF (65–72). Specifically, Pakhomov's group recently published that 200 ns pulses can cause a 1,000-fold reduction of tissue stimulation compared to 100 µs pulses (72).

In this study, we measured MRSA inactivation by nsPEF alone and in combination with antibiotics approved to treat SSTIs, namely vancomycin, doxycycline, and daptomycin. Previous studies were limited to testing the bacterial susceptibility to various antibiotics only after nsPEF were delivered (57–59). Here, we directly compared the efficacy of multiple antibiotics administered either before or after the pulse treatment. Our results show that treatment duration and order (nsPEF-antibiotics and antibiotics-nsPEF) are essential in determining the most effective result, findings that will guide experimental design for future *in vivo* studies of the interaction between nsPEF and antibiotics to treat SSTIs.

## RESULTS

### Sensitivity of planktonic MRSA to nsPEF treatments

Our initial experiments sought to establish optimal laboratory growth conditions for the planned experiments. MRSA growth in liquid culture was assessed in different growth media commonly used to culture MRSA (data not shown). We found that MRSA grows rapidly and consistently in LB broth and reaches a stationary optical density of 1.2–1.4 within 12 h (Fig. 1A). Twenty-four hours was sufficient time for the growth of robust MRSA biofilms on plastic surfaces (Fig. 1B).

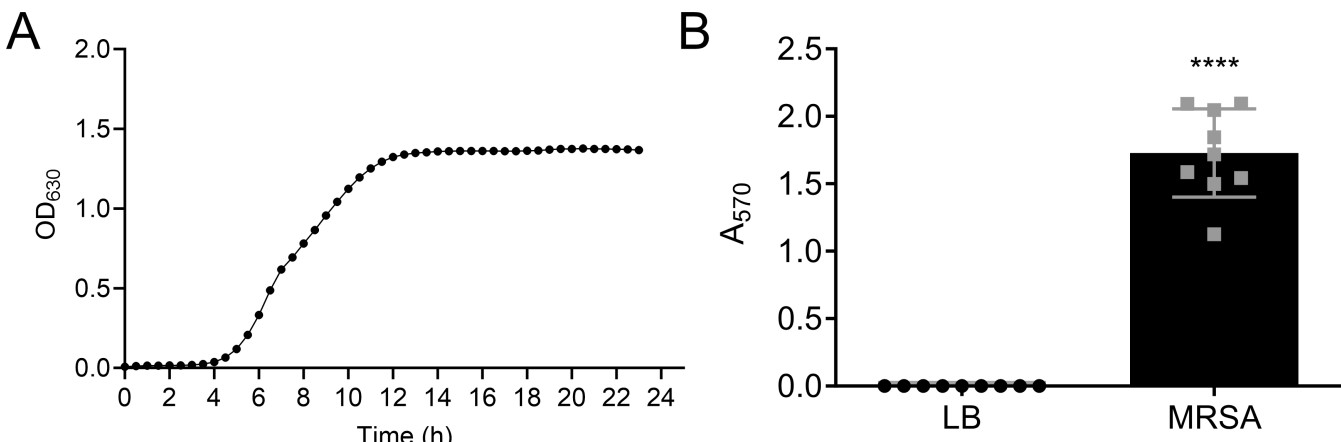

FIG 1   *S. aureus* growth in LB broth. (A) Planktonic MRSA cultures were diluted using a 1:20 ratio in LB broth in sterile 96-well plates. Plates were incubated at 37.0°C for 24 h in a microplate reader, which was set to constantly shake at medium intensity, and the optical density was recorded at 630 nm every 30 min. (B) Planktonic MRSA cultures were diluted 1:10 in LB broth in sterile 12-well plates. Plates were incubated at 37.0°C for 48 h before crystal violet staining was used to measure biofilm production. Shown are the means and standard deviations of nine samples. ****$P < 0.0001$.

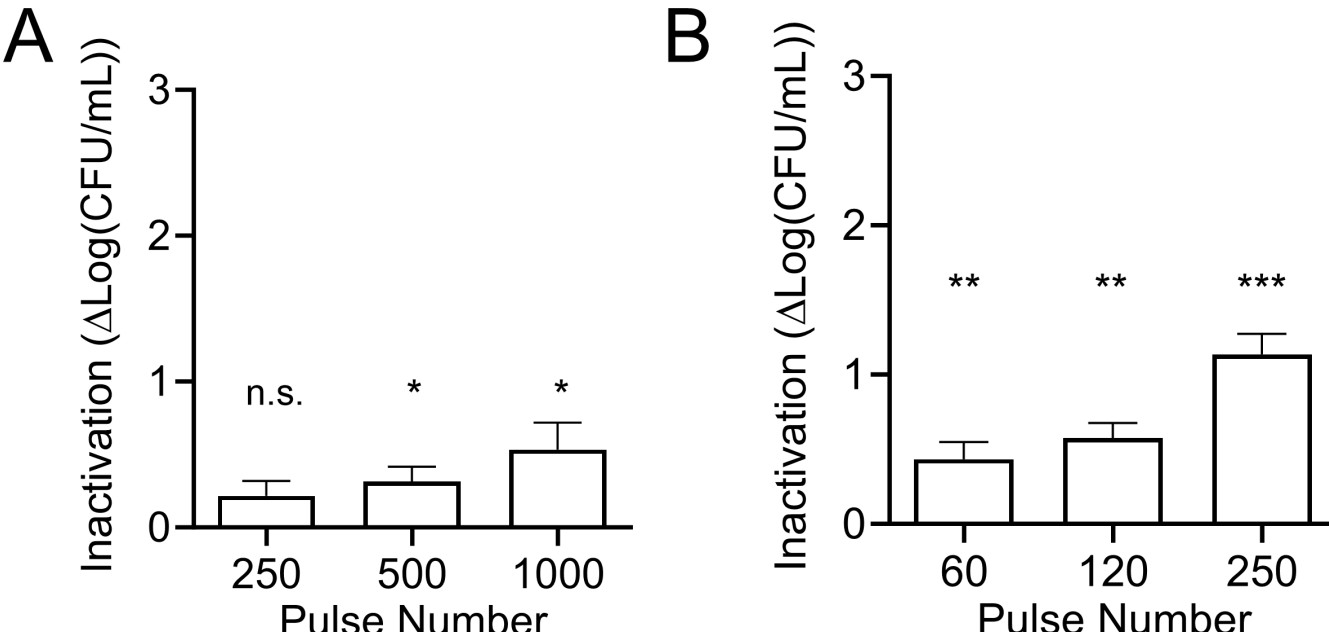

**FIG 2** Effect of nsPEF on planktonic *S. aureus* viability. Cells in the exponential phase were treated with the indicated numbers of either (A) 300 or (B) 600 ns pulses. All pulses were 28 kV/cm, 1 Hz. Treated samples were plated in 10-fold dilutions on LB agar plates and colony-forming units were quantified 40 h post-treatment. Inactivation is quantified as $\log(CFU/mL)_{sham}$ − $\log(CFU/mL)_{nsPEF}$ and is individually calculated for each sample. Shown are the means and standard deviations of at least three independent samples. n.s., not significant. * $P < 0.05$, **$P < 0.01$, and ***$P < 0.001$, one-sample *t*-test for the difference from untreated control (0).

Inactivation by nsPEF of planktonic MRSA in the exponential growth phase was initially measured using two different pulse durations, namely 300 ns, the shortest duration our generator can produce, and 600 ns (0–1,000 pulses, 28 kV/cm, 1 Hz). Our results show that while MRSA was moderately affected by both pulse durations (Fig. 2), 600 ns pulses were more efficient. Specifically, 250 pulses of 300 ns each caused a very modest 0.2 $\log_{10}$ reduction in viability, and increasing the number of pulses to 500 or 1,000 only increased the killing effect to 0.3 and 0.5 $\log_{10}$ reductions, respectively (Fig. 2A). Meanwhile, 60 pulses of 600 ns each caused a 0.4 $\log_{10}$ reduction in viability and increasing the number of pulses to 120 or 250 pulses increased the killing effect to 0.5 and 1.1 $\log_{10}$ reductions, respectively (Fig. 2B). PEF treatments are intended to be a non-thermal method to inactivate microorganisms. However, it is well-known that an increase in temperature due to Joule heating can be associated with high pulse doses. Table 1 shows that the highest 600-ns pulse dose used in this study (250 pulses) increased the sample temperature from 24.4°C ± 0.3°C to 38.7°C ± 0.3°C, while 60 and 120 pulses increased the temperature to 28.6°C ± 0.1°C and 33.1°C ± 0.2°C, respectively. Although *S. aureus* can grow over a wide range of temperatures (6.5°C–46°C) with an optimal range between 30°C and 37°C (73), we could not exclude the possibility that a rapid 14°C rise in temperature during treatment could affect the electroporation phenomenon and/or initiate stress responses. Therefore, to minimize the effect of heating, the condition utilizing 250 pulses at 600 ns pulse duration was discontinued so that all data would be collected within the optimal temperature range for *S. aureus* growth. All subsequent experiments used 60 or 120 pulses with the 600 ns pulse duration, as indicated.

## nsPEF pre-treatment or post-treatment increases the antimicrobial effect of limited daptomycin exposure

Next, we asked whether nsPEF treatments could sensitize MRSA to transient exposure to the SSTI-approved antibiotic daptomycin. Daptomycin is a lipophilic peptide that inserts

**TABLE 1** Temperatures measured immediately after the delivery of the indicated number of 600 ns pulses (28 kV/cm, 1 Hz)

| No. of pulses | Temperature (℃) |
| --- | --- |
| 0 | 24.4 ± 0.3 |
| 60 | 28.6 ± 0.1 |
| 120 | 33.1 ± 0.2 |
| 250 | 38.7 ± 0.3 |

into the bacterial cell membrane, causing rapid membrane depolarization and potassium ion efflux (74). We found that 4 µg/mL daptomycin inhibits MRSA growth for at least 16 h, while lower doses slow growth kinetics slightly but do not prevent cultures from reaching maximal density within 16 h (Fig. S1A). We hypothesized that the membrane defects created by daptomycin would increase the efficacy of nsPEF treatment. To test this hypothesis, we first measured the ability of daptomycin to reduce the number of viable cells in exponentially growing MRSA cultures. As expected, daptomycin exposure reduced the number of colony-forming units per milliliter in a dose-dependent manner, with a 90-min exposure to 8 µg/mL daptomycin, the highest concentration tested, leading to 3.3 $\log_{10}$ reduction in viability (Fig. 3A). Next, we measured the effect of combining daptomycin with nsPEF (Fig. 3B). Cells were preincubated for 90 min with different sub-lethal doses of the antibiotic (0.5, 1, and 2 µg/mL), then treated with nsPEF (0, 60, and 120 pulses 600 ns, 28 kV/cm, 1 Hz), and immediately plated on LB agar plates without antibiotic (for a schematic diagram of the experimental workflow see Fig. S2A). Our results show that pretreatment with daptomycin significantly increased MRSA sensitivity to nsPEF (Fig. 3B). Combining 2 µg/mL of daptomycin with nsPEF caused nearly 3 $\log_{10}$ reduction, comparable to the effect of 8 µg/mL of antibiotic alone. Similarly, at all of the antibiotic concentrations tested, antibiotics combined with pulses reduced culture viability significantly more than pulses alone (Fig. 3B). Next, we investigated whether the order in which the combined treatments were applied affected our results. Samples were either pretreated with 0.5 µg/mL of daptomycin for 90 min and then exposed to the nsPEF (600 ns, 28 kV/cm, 1 Hz) or exposed to nsPEF and then incubated with the antibiotic for 90 min. Our results show that regardless of the administration order, the co-treatment inactivates MRSA culture more strongly than either daptomycin or nsPEF alone (Fig. 4). Each of the monotherapies reduced culture viability by less than 1 $\log_{10}$, while prior exposure to daptomycin allowed 60 or 120 pulses to reduce viability by 1.5 and 1.9 $\log_{10}$, respectively. Pretreatment with 60 or 120 pulses sensitized the cells to daptomycin, allowing culture inactivation by 2.0 and 2.3 $\log_{10}$, respectively (Fig. 4).

## Only nsPEF pre-treatment sensitizes MRSA to doxycycline

Next, we asked whether nsPEF treatment would also enhance the effects of antibiotics with different mechanisms of action. Doxycycline inhibits bacterial protein synthesis by reversibly binding to the 30S ribosomal subunits, blocking the binding of the aminoacyl tRNA to the mRNA (75, 76). We found that 2 µg/mL doxycycline partially suppressed MRSA growth over 16 h, while 4 µg/mL treatment completely suppressed it (Fig. S1B). Unlike daptomycin, doxycycline must enter the cell to have an effect; the thick cell wall of Gram-positive pathogens such as MRSA can impede this. We therefore hypothesize that the damage created by nsPEF could increase MRSA permeability to doxycycline. As with daptomycin, we first measured the sensitivity of exponentially growing MRSA to doxycycline monotherapy. Interestingly, all tested doses (0–32 µg/mL) only mildly affected MRSA viability, suggesting that the 90-min contact time was not sufficient to cause significant bactericidal effects (Fig. 5A). Next, we measured the effect of combining doxycycline with nsPEF. Cells were incubated for 90 min with 4 µg/mL doxycycline either before or after treatment with nsPEF (0, 60, and 120 pulses 600 ns, 28 kV/cm, 1 Hz) and immediately plated on LB plates without antibiotic. Our results show that pretreating MRSA with doxycycline did not increase cells' sensitivity to nsPEF, while nsPEF

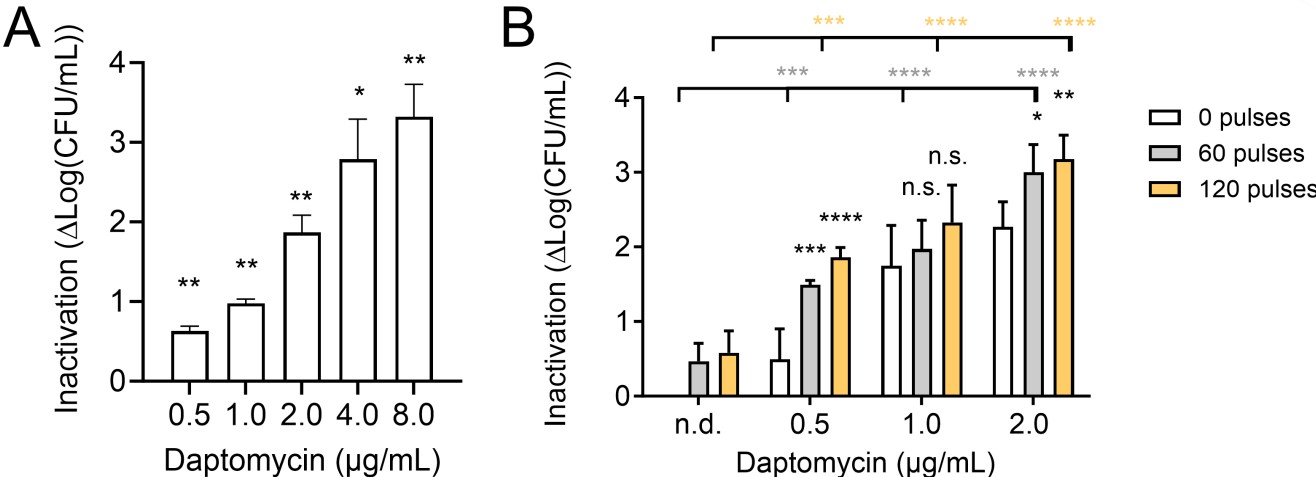

**FIG 3** Pretreatment with daptomycin increases nsPEF cytotoxic effect. (A) Inactivation of exponentially growing MRSA cultures by 90 min of incubation with the indicated doses of daptomycin. Inactivation is quantified as $\log(CFU/mL)_{untreated} - \log(CFU/mL)_{daptomycin}$ and is individually calculated for each sample. Treated samples are compared to 0 (untreated control) by one-sample $t$-test. Numbers shown represent the means and standard deviations of at least three independent samples. (B) Inactivation of samples treated with the indicated concentrations of daptomycin for 90 min before exposure to 0, 60, or 120 pulses (600 ns, 28 kV/cm, 1 Hz). Inactivation is quantified as $\log(CFU/mL)_{untreated} - \log(CFU/mL)_{treated}$ and is individually calculated for each sample. Black asterisks: samples treated with daptomycin and pulses are compared to those treated with daptomycin alone by one-way ANOVA. Gray asterisks: samples treated with daptomycin and 60 pulses are compared to those treated with pulses alone by one-way ANOVA. Orange asterisks: samples treated with daptomycin and 120 pulses are compared to those treated with pulses alone by one-way ANOVA. Numbers shown represent the means and standard deviations of four independent samples. n.d., no drug; n.s., not significant; *$P < 0.05$; **$P < 0.01$; ***$P < 0.001$; and ****$P < 0.0001$.

significantly potentiated MRSA susceptibility to consequent doxycycline incubation (Fig. 5B). Pre-treatment with doxycycline caused a small but significant enhancement to the effects of pulses, increasing the inactivation effect from 0.4 $\log_{10}$ in the presence of doxycycline alone to 0.6 when doxycycline is followed by 60 or 120 pulses. Pulses sensitize MRSA to doxycycline much more strongly, as pre-treatment with 60 pulses increases culture inactivation by doxycycline from 0.6 to 1.3, and pre-treatment with 120 pulses increases it further to 1.5 (Fig. 5B). While the effect of the pulses on cells previously exposed to doxycycline was not significant compared to that of the pulses alone, pulses followed by doxycycline exposure are much more lethal to the MRSA cultures than pulses alone (Fig. 5B). These results suggest that cell permeabilization by nsPEF enhances doxycycline penetration into the bacterial cytoplasm, while pretreatment with the antibiotic has no effect on the cellular response to nsPEF.

## nsPEF does not strongly affect the efficacy of a transient exposure to vancomycin

Vancomycin is a glycopeptide antibiotic that exerts its bactericidal effect by inhibiting the polymerization of peptidoglycans in the bacterial cell wall and is broadly effective against Gram-positive bacteria (77). Vancomycin is often clinically used to treat MRSA infections (78). The vancomycin MIC90 for MRSA has previously been reported as 2 µg/mL (79). We found that 2 µg/mL vancomycin partially suppressed MRSA growth for 16 h, with complete inhibition at 4 µg/mL (Fig. S1C). We hypothesized that the destabilization of the cell wall by vancomycin could enhance osmotic cell swelling and consequent cell death in electroporated MRSA. Like doxycycline, incubation with all tested doses of vancomycin for 90 min was not enough to measure substantial cytotoxic effects against exponentially growing cultures (Fig. 6A). Treatment with 4 µg/mL of vancomycin followed by treatment with nsPEF (0, 60, and 120 pulses 600 ns, 28 kV/cm, 1 Hz) had no significant effect on bacterial inactivation, while nsPEF treatment with 60 pulses followed by antibiotic exposure modestly increased inactivation from 1.3 to 1.9 $\log_{10}$ for 60 pulses

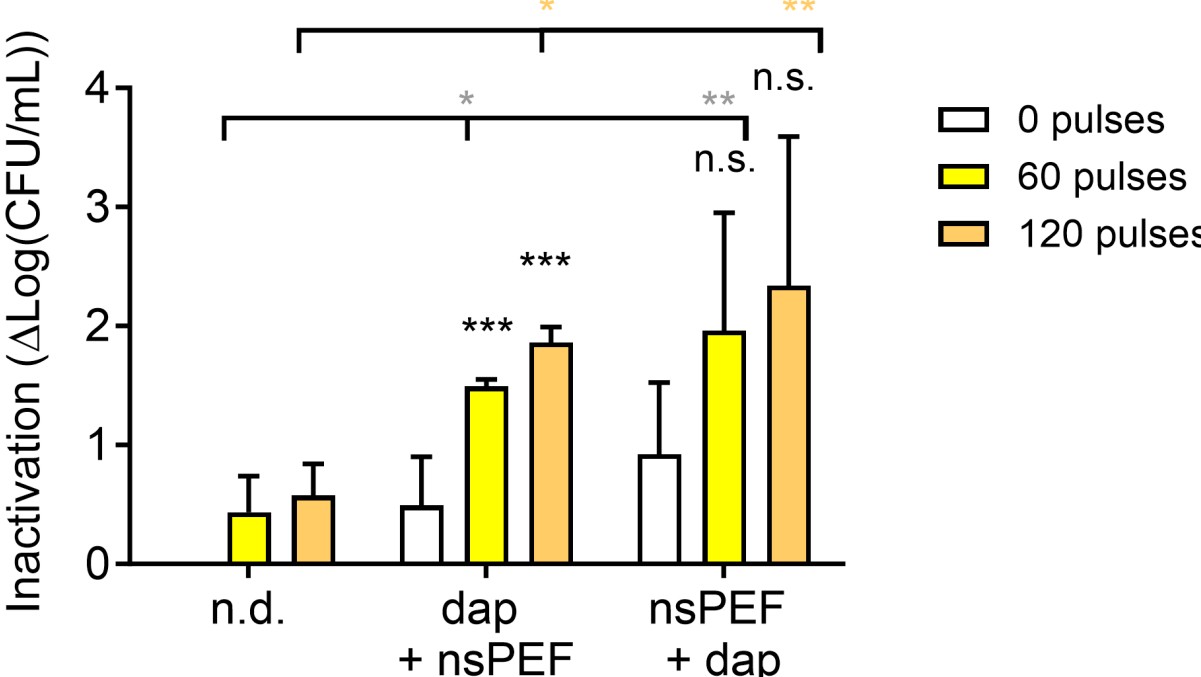

**FIG 4** Daptomycin and nsPEF mutually enhance each other regardless of the order of application. Exponentially growing MRSA cultures were either pre-incubated with 0.5 µg/mL daptomycin for 90 min and then exposed to 0, 60, and 120 pulses (Dap + nsPEF) or exposed to nsPEF and then incubated with the antibiotic (nsPEF +Dap). Inactivation is quantified as $\log(CFU/mL)_{untreated} - \log(CFU/mL)_{treated}$. Black asterisks: samples treated with daptomycin and pulses are compared to those treated with daptomycin alone by one-way ANOVA. Gray asterisks: samples treated with daptomycin and 60 pulses are compared to those treated with pulses alone by one-way ANOVA. Orange asterisks: samples treated with daptomycin and 120 pulses are compared to those treated with pulses alone by one-way ANOVA. Numbers shown represent the means and standard deviations of at least four independent samples. n.s., not significant; *$P < 0.05$; **$P < 0.01$; and ***$P < 0.001$.

(Fig. 6B). Treatment with 120 pulses increased inactivation by vancomycin from 1.6 to 2.1 $\log_{10}$, but the results were not statistically significant (Fig. 6B). Pre-treatment with 60 pulses followed by vancomycin exposure was the only combinatorial therapy to exhibit any significant enhancement over nsPEF monotherapy (Fig. 6B). Under our experimental conditions, it appears that antibiotic destabilization of the cell wall does not exhibit the same mutual enhancement with nsPEF treatment as antibiotic destabilization of the cell membrane.

### nsPEF treatment sensitizes cells to prolonged antibiotic exposure

Because both doxycycline and vancomycin alone had only modest effects on MRSA viability after 90 min of exposure, we decided to test the effect of increased contact time with these antibiotics. To do this, we employed dilution plating on plates containing these antibiotics. After nsPEF treatment (0, 60, and 120 pulses 600 ns, 28 kV/cm, 1 Hz), 10-fold serial dilutions of treated and untreated cells were spotted onto LB plates containing antibiotics (Fig. 7). In samples with reduced viability, fewer dilutions are needed before no visible growth is detected after 24 h. Both doxycycline and vancomycin visibly reduced MRSA cell density on a serial dilution plate (compare yellow rectangles in Fig. 7A and C). The effect of nsPEF treatment alone was apparent on the dilution plates in the absence of antibiotics (Fig. 7A and C, see blue rectangles). However, the difference between nsPEF-treated and untreated cells was much more pronounced on plates containing antibiotics (Fig. 7A and C, red rectangles). The trends observed on the doxycycline plates were not statistically significant (Fig. 7B), but nsPEF combined with sustained vancomycin exposure inactivated MRSA significantly more than either treatment alone (Fig. 7D).

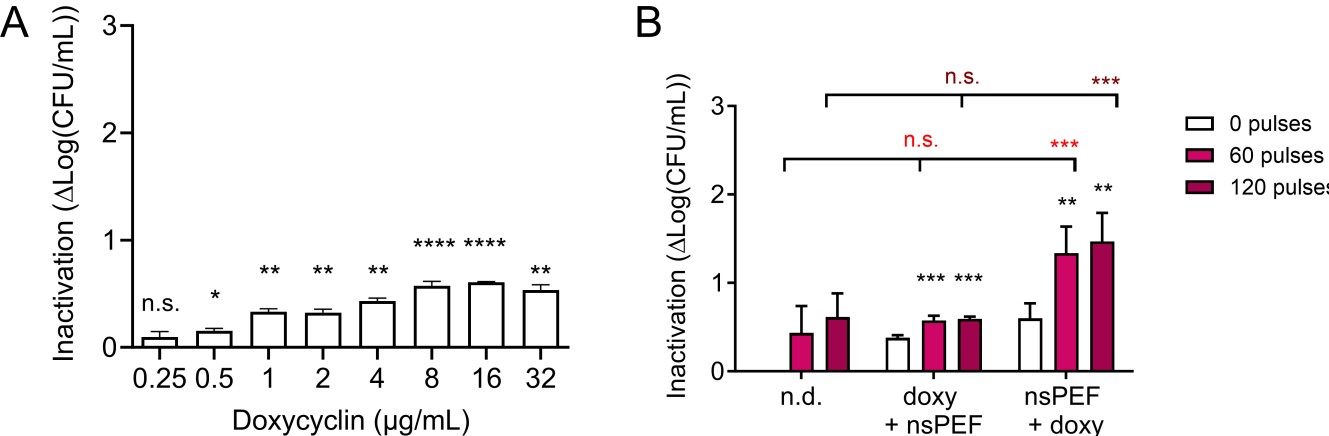

**FIG 5** Pre-treatment with doxycycline does not increase nsPEF efficacy but nsPEF sensitizes MRSA to doxycycline. (A) Inactivation of exponentially growing MRSA cultures by 90 min of incubation with the indicated doses of doxycycline. Inactivation is quantified as $\log(CFU/mL)_{untreated} - \log(CFU/mL)_{treated}$ and is individually calculated for each sample. Treated samples are compared to 0 (untreated control) by one-sample $t$-test. Numbers shown represent the means and standard deviations of at least three independent samples. (B) Inactivation of exponentially growing MRSA cultures either pre-incubated with 4 µg/mL doxycycline for 90 min and then exposed to 0, 60, and 120 pulses (doxy + nsPEF) or exposed to nsPEF and then incubated with the antibiotic (nsPEF + doxy). Inactivation is quantified as $\log(CFU/mL)_{untreated} - \log(CFU/mL)_{treated}$. Black asterisks: samples treated with doxycycline and pulses are compared to those treated with doxycycline alone by one-way ANOVA. Light fuchsia symbols: samples treated with doxycycline and 60 pulses are compared to those treated with pulses alone by one-way ANOVA. Dark fuchsia symbols: samples treated with doxycycline and 120 pulses are compared to those treated with pulses alone by one-way ANOVA. Numbers shown represent the means and standard deviations of at least three independent samples. n.s., not significant. *$P < 0.05$, **$P < 0.01$, ***$P < 0.001$, and ****$P < 0.0001$.

Plating nsPEF-treated MRSA on daptomycin plates led to inconclusive results as prolonged incubation of cells with this antibiotic caused either no effect or complete growth inhibition at all tested concentrations (Fig. S3).

## Genetic resistance to methicillin is not affected by nsPEF treatment

Next, we asked whether treatment with nsPEF affected MRSA resistance to methicillin. Cells were exposed to nsPEF (0, 60, and 120 pulses 600 ns, 28 kV/cm, 1 Hz) and immediately plated on plates containing 32 µg/mL methicillin, a concentration which can slow but not prevent MRSA proliferation (Fig. S1D). We found that methicillin had no impact on MRSA culture inactivation by nsPEF (Fig. 8). These results can be explained by the fact that methicillin resistance depends on cytoplasmic penicillin-binding proteins with reduced affinity for methicillin (80), a resistance mechanism which is unlikely to be modulated by nsPEF-induced plasma membrane perturbation.

## Effect of nsPEF/antibiotics combined treatment of MRSA biofilm viability

Because *S. aureus* living within biofilms is more resistant to antibiotics than planktonic bacteria (81), we investigated the effect of combining nsPEF with either doxycycline or vancomycin on MRSA biofilms' viability. Biofilms were washed and adherent cells were manually scraped from the plastic growth surface and resuspended into sterile LB broth. Cells were dispersed by vortexing and aliquoted into electroporation cuvettes. After nsPEF treatment (0, 60, and 120 pulses 600 ns, 28 kV/cm, 1 Hz), samples were plated on LB plates containing 1 µg/mL of either doxycycline or vancomycin and colonies were counted in 40 h (Fig. 9). nsPEF treatment alone had very little effect on biofilm-grown cells, with 60 and 120 pulses reducing culture viability by less than 0.3 $\log_{10}$ (Fig. 9). Doxycycline alone only reduced culture viability of 0.4 $\log_{10}$, while pre-treatment with 60 or 120 pulses allowed doxycycline to impact the biofilm-grown cells by 0.5 and 0.6 $\log_{10}$ (Fig. 9A). The effects of the co-treatment were significantly enhanced compared to either treatment alone, but their overall impact was modest. Vancomycin treatment had more of an impact on the biofilm-grown cells, reducing viability by 1.1 $\log_{10}$. Pre-treatment

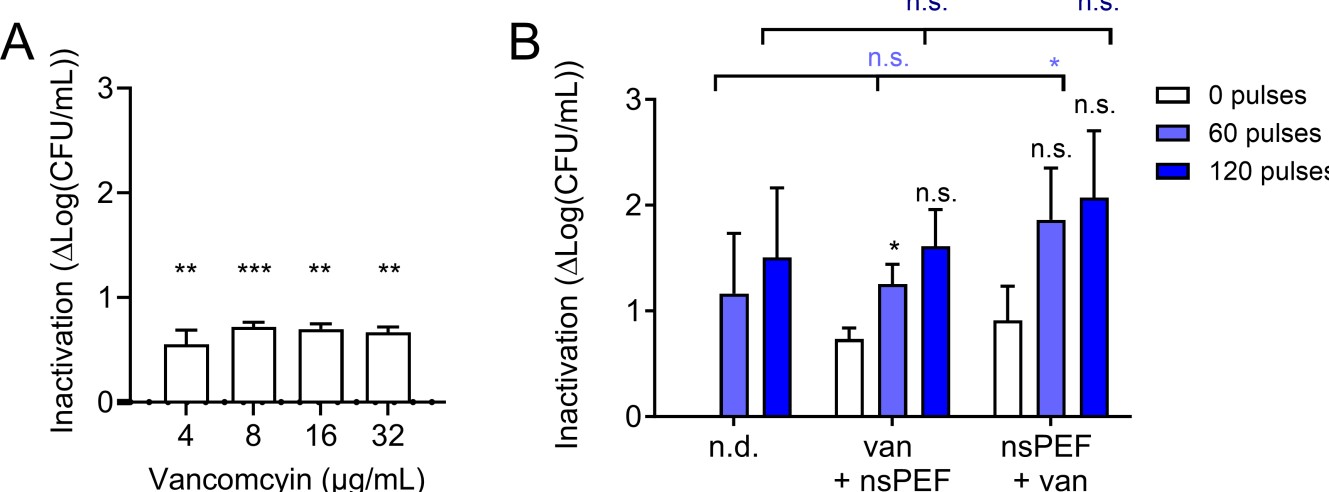

**FIG 6** Vancomycin and nsPEF treatment do not strongly enhance each other. (A) Inactivation of exponentially growing MRSA cultures by 90 min of incubation with the indicated doses of vancomycin. Inactivation is quantified as $log(CFU/mL)_{untreated} - log(CFU/mL)_{treated}$ and is individually calculated for each sample. Treated samples are compared to 0 (untreated control) by one-sample $t$-test. Numbers shown represent the means and standard deviations of at least three independent samples. (B) Inactivation of exponentially growing MRSA cultures either pre-incubated with 4 µg/mL vancomycin for 90 min and then exposed to 0, 60, and 120 pulses (van + nsPEF) or exposed to nsPEF and then incubated with the antibiotic (nsPEF +van). Inactivation is quantified as $log(CFU/mL)_{untreated} - log(CFU/mL)_{treated}$. Black symbols: samples treated with vancomycin and pulses are compared to those treated with vancomycin alone by one-way ANOVA. Light blue symbols: samples treated with vancomycin and 60 pulses are compared to those treated with 60 pulses alone by one-way ANOVA. Dark blue symbols: samples treated with vancomycin and 120 pulses are compared to those treated with 120 pulses alone by one-way ANOVA. Numbers shown represent the means and standard deviations of at least three independent samples. n.s., not significant; $* P < 0.05$; $**P < 0.01$; and $***P < 0.001$.

with 60 or 120 nsPEF pulses increased this effect to 1.8 and 1.9 $log_{10}$ (Fig. 9B). nsPEF enhancement of vancomycin treatment alone did not achieve statistical significance, but did trend upward, suggesting that investigation of vancomycin penetrance and efficacy in intact biofilms subjected to nsPEF treatment could be a promising area of future investigation.

## MATERIALS AND METHODS

### Bacterial strains and growth conditions

All assays were conducted with *S. aureus* Xen 31 MRSA strain (Perkin Elmer, Waltham, MA, USA). Bacterial colonies were maintained on LB plates containing 17 g/L agar (Fisher Scientific, Waltham, MA, USA). Bacteria were grown in Luria Bertani Miller (LB-Miller; Fisher Scientific) broth at 37°C until they reached exponential phase (optical density of 0.4–0.7 at 600 nm) in a shaking incubator (New Brunswick Scientific, Edison, NJ, USA) at 250 rpm. Bacterial growth was measured using the DU 730 spectrophotometer (Beckman Coulter, Inc., Chaska, MN, USA).

### Growth curves

MRSA cultures were prepared by inoculating single colonies into 2 mL of LB broth and grown for 16 h at 37°C with constant shaking at 250 rpm. The overnight cultures were diluted 1:20 into fresh LB broth, grown until they reached the exponential phase ($OD_{600nm}$: 0.4–0.7), and diluted 1:10 into a sterile 96-well plate (Fisher Scientific) containing a range of antibiotic concentrations in LB broth. The antibiotics that were used were: vancomycin (VWR, Suwanee, GA, USA), doxycycline (Cayman Chemicals, Ann Arbor, MI, USA), daptomycin (Fisher Scientific), and methicillin (Fisher Scientific). Plates were incubated at 37.0°C for 24 h in a BioTek synergy microplate reader, which was set to constantly shake at medium intensity and absorbance was recorded at 630 nm every 30 min.

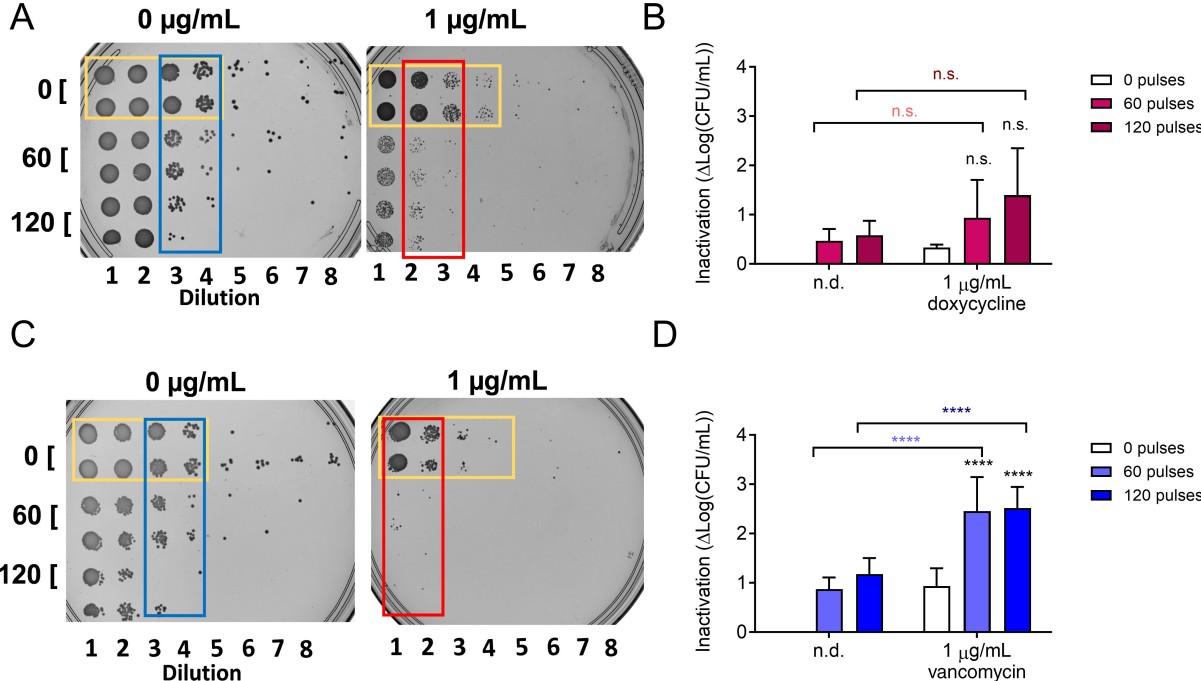

**FIG 7** Effect of nsPEF and extended incubation with doxycycline (A) and vancomycin (B) on MRSA viability. In panels A and C, a replica plating device was used to reproducibly spot 10-fold serial dilutions onto agar plates. Shown are representative images of eight serial 10-fold dilutions of exponentially growing MRSA treated with 0, 60, or 120 pulses (600 ns, 28 kV/cm, 1 Hz) and then plated either on control LB plates (A and C left images) or plates containing 1 µg/mL doxycycline (A, right image) or vancomycin (C, right image). Yellow and blue rectangles highlight the effect of the monotreatment with antibiotics and nsPEF, respectively. Red rectangles show the effect of the combined treatment (see text for details). In panels B and D, a quantification of the effects seen in panels A and C was done for one optimal dilution. Inactivation is quantified as log(CFU/mL)$_{untreated}$ – log(CFU/mL)$_{treated}$. Black symbols: samples treated with antibiotics and pulses are compared to those treated with antibiotics alone by one-way ANOVA. Light fuchsia symbols: samples treated with daptomycin and 60 pulses are compared to those treated with pulses alone by one-way ANOVA. Dark fuchsia symbols: samples treated with daptomycin and 120 pulses are compared to those treated with pulses alone by one-way ANOVA. Light blue asterisks: samples treated with vancomycin and 60 pulses are compared to those treated with 60 pulses alone by one-way ANOVA. Dark blue asterisks: samples treated with vancomycin and 120 pulses are compared to those treated with 120 pulses alone by one-way ANOVA. Data shown represent the mean and standard deviation of at least three samples. n.s., not significant and **** $P < 0.0001$.

## Pulsed electric field exposure methods

MRSA cultures were prepared by inoculating single colonies into 2 mL of LB broth and grown for 18 h at 37°C with constant shaking. Starter cultures were diluted 1:20 into LB broth to reach the exponential phase (OD$_{600}$ = 0.4–0.7), and 90 µL samples of this suspension were loaded into 1 mm gap electroporation cuvettes (BioSmith, Vandergrift, PA, USA). Samples in electroporation cuvettes were exposed to nsPEF in LB broth with a conductivity of 1.73 S/m at room temperature. Trapezoidal pulses of 300 or 600 ns duration were produced by a CellFX generator (Pulse Biosciences Inc, Hayward, CA, USA). The output stage of the pulse generator was optimized for the 10-ohm impedance presented by the cell suspensions in a 1 mm electroporation cuvette. The pulse amplitude and shape were monitored at the cuvette using a LeCroy Wave-Surfer 3034z oscilloscope (Teledyne Lecroy, Chestnut Ridge, NY, USA). A picture of the experimental setup and the waveform are shown in Fig. S2B and C. Temperature changes were measured immediately after nsPEF using a thermocouple thermometer (Physitemp, Clifton, NJ, USA). The nsPEF-treated cells underwent a serial dilution in LB broth before being plated on LB agar plates. The plates were incubated at 37°C for 40 h before the number of colonies was counted. All experiments included an untreated "sham" control that was prepared the same way as the experimental sample but not subjected to nsPEF.

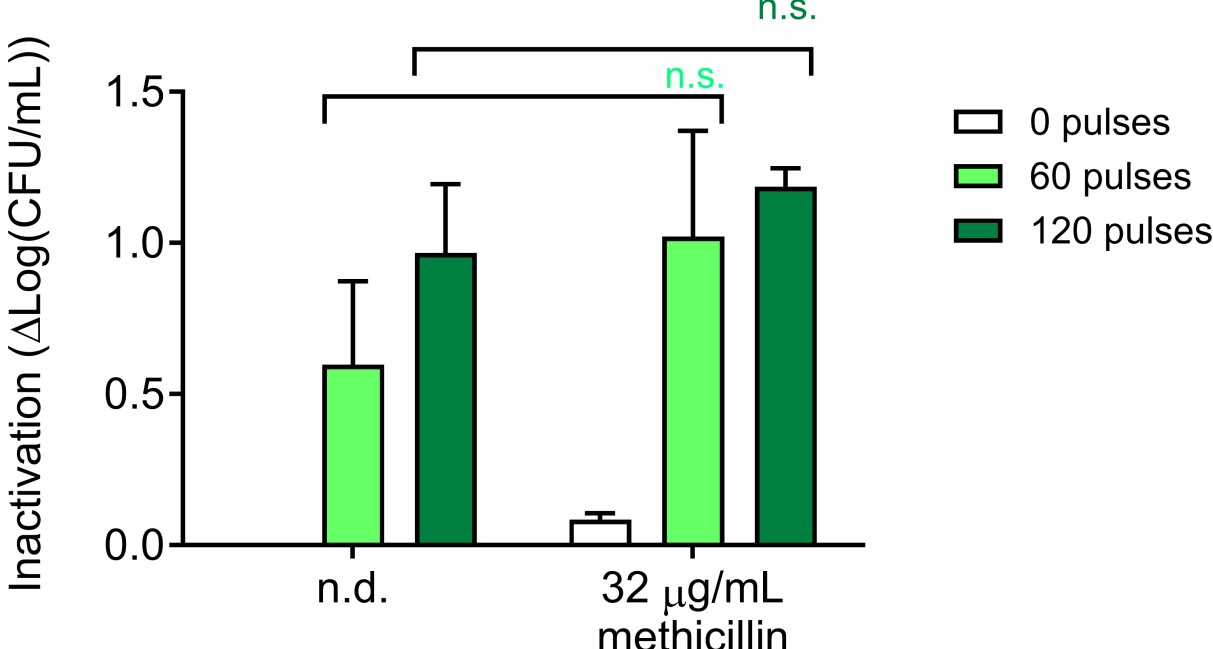

**FIG 8** Treatment with nsPEF does not sensitize MRSA to methicillin. Inactivation of exponentially growing MRSA cultures exposed to 0, 60, and 120 pulses (600 ns, 28 kV/cm, 1 Hz) and then incubated with 32 µg/mL methicillin for 40 h. Inactivation is quantitated as $\log(CFU/mL)_{untreated} - \log(CFU/mL)_{treated}$. Samples treated with methicillin and pulses are compared to those treated with pulses alone by one-way ANOVA. Numbers shown represent the means and standard deviations of at least three independent samples. n.s., not significant.

## Antibiotic treatments

A schematic diagram of the experimental workflow is shown in Fig. S2A. For short antibiotic incubations either pre- or post-nsPEF treatment, MRSA cultures in the exponential growth phase ($OD_{600}$ = 0.4–0.7) were treated with a range of concentrations for each of the antibiotics: 0–8 µg/mL daptomycin, 0–32 µg/mL doxycycline, and 0–32 µg/mL vancomycin. Antibiotic exposure lasted 90 min at 37°C with constant shaking at 250 rpm. For prolonged antibiotic exposures, samples were exposed to nsPEF as described above, diluted, and spread on LB-agar plates containing the indicated concentrations of antibiotics. Plates were incubated at 37°C for 40 h before the number of colonies was counted.

## Replicate plating

Exponential phase cultures were exposed to 600 ns pulses (0, 60, and 120 pulses) and serially diluted in a sterile 96-well plate (ThermoFisher Scientific). A sterile replica plater ("frogger") for a 96-well plate (Sigma Aldrich) was used to stamp the desired samples onto LB ± antibiotic agar plates. The plates were incubated at 37°C for 24 h before being scanned on a ChemiDoc MP Imaging System (BioRad).

## Biofilm formation, visualization, and quantification

A 12-well tissue culture-treated plate (Fisher Scientific) containing 1.8 mL of fresh LB broth in each well was inoculated with 200 µL of MRSA exponential phase cultures. The plate was then incubated at 37°C for 24 h. To quantify the biofilms, the supernatant from each well was removed by pipetting, and the adhered biofilms were washed with 1 mL of 1× phosphate-buffered saline (PBS) solution. The washed biofilms were then stained for 30 min with 1 mL of 0.1% crystal violet (Sigma Aldrich) and washed two additional times with 1× PBS. The adhered and stained biofilms were suspended in 70% ethanol. The plate was placed in the Bio-Tek synergy microplate reader (Marshall Scientific), which recorded the $OD_{570}$, after shaking the plate at medium intensity.

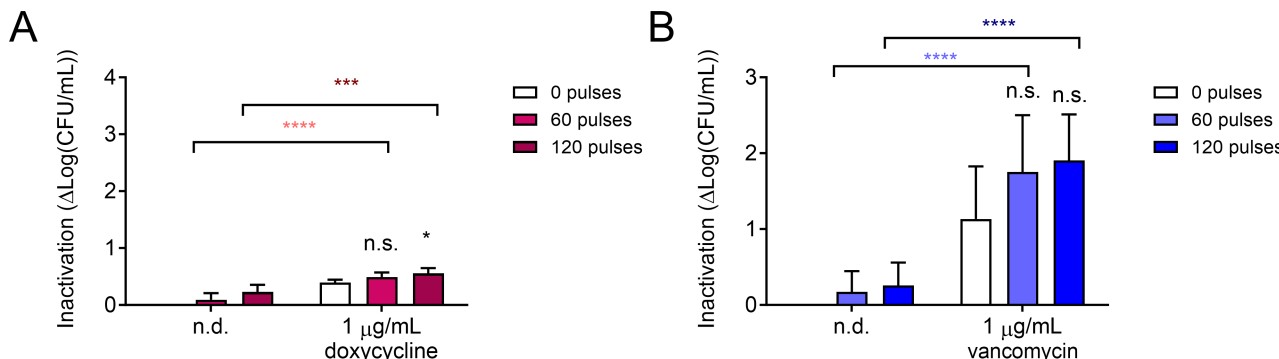

**FIG 9** Effect of nsPEF on antibiotic susceptibility of biofilm-derived cells. MRSA cells scraped out of biofilms were homogenized in solution and treated with 0, 60, or 120 pulses before serial dilution plating on either control LB plates or plates containing 1 µg/mL doxycycline (A) or 1 µg/mL vancomycin (B). Inactivation is quantified as $\log(CFU/mL)_{untreated} - \log(CFU/mL)_{treated}$ and is individually calculated for each sample. Black asterisk: samples treated with antibiotics and pulses are compared to those treated with antibiotics alone by one-way ANOVA. Light fuchsia asterisks: samples treated with doxycycline and 60 pulses are compared to those treated with pulses alone by one-way ANOVA. Dark fuchsia asterisks: samples treated with doxycycline and 120 pulses are compared to those treated with pulses alone by one-way ANOVA. Light blue asterisks: samples treated with vancomycin and 60 pulses are compared to those treated with 60 pulses alone by one-way ANOVA. Dark blue asterisks: samples treated with vancomycin and 120 pulses are compared to those treated with 120 pulses alone by one-way ANOVA. Data represent the means and standard deviations of at least five independent samples. n.s., not significant and $*P < 0.05$.

## nsPEF and antibiotic treatment of biofilm-derived cells

To treat the biofilms, the supernatant was removed, biofilms were washed with 1× PBS, and adherent cells were manually scraped off the plastic surface with pipette tips and suspended in 1 mL of fresh LB broth. The sample was then vortexed to disrupt the biofilm structure. A 90-µL aliquot of this sample was transferred to an electroporation cuvette for nsPEF as described above. After the nsPEF treatment, the sample was plated on LB ± antibiotic (1 µg/mL doxycycline or 1 µg/mL vancomycin) agar plates. The plates were incubated at 37°C for 40 h before the number of colonies was counted.

## Determination of inactivation rates

Immediately after pulsing, 900 µL of LB broth was added to each cuvette and mixed by pipetting. The resulting 1 mL samples were serially diluted up to $10^{-7}$, 100 µL of each sample was plated on duplicate LB agar plates, and colonies were counted after 40-h incubation at 37°C. Only counts between 0 and 300 CFU per plate were considered. Inactivation rates are expressed as $\log_{10}(CFU/mL_{sham} - CFU/mL_{treated})$.

## Statistical analyses

Data are presented as mean ± SD for *n* independent experiments. Statistical calculations, including data fits, and data plotting were accomplished using Prism (GraphPad). All quantitative experiments were performed in duplicate and repeated a minimum of three times.

## DISCUSSION

*S. aureus* is a natural component of the commensal skin microbiota but can become an opportunistic pathogen when the skin, the first line of immune defense, is breached (82–84). Treatment of the resulting SSTI is complicated by the high prevalence of methicillin resistance among *S. aureus* strains (85). Many *S. aureus* infections, both methicillin-sensitive and methicillin-resistant, are currently treated using doxycycline, vancomycin, or daptomycin, alone or in tandem with physical debridement methods (86). However, resistance has also developed to these antibiotics (87–89). Effective methods of treatment are necessary to reduce the burden of *S. aureus* in healthcare settings (90).

This study makes use of nsPEF in combination with antibiotic treatments to inactivate MRSA. Pulsed electric fields used for bacterial inactivation are traditionally of microsecond duration and have been seen to impact bacterial viability with a range of pulse amplitudes (46, 91). Numerous studies have demonstrated that bacteria, when exposed to PEF, show both membrane and cell wall damage as well as subsequent cell death (92–94). Additionally, resistance to this treatment type has not been detected before 30 generations (95). While these studies are promising, few have investigated the synergistic effects of shorter nanosecond duration pulses in combination with antibiotics (58).

Our study shows that MRSA in the exponential growth phase is mildly inactivated when treated with 300 or 600 nanosecond pulses with an electric field strength of 28 kV/cm, the maximum field we could reach with our setup. The inactivation observed is in line with previously completed studies on other Gram-positive bacterial species (57, 58). A previous study investigating the effects of 300 ns pulses on *S. aureus* viability showed an inactivation amount of 0.2 $\log_{10}$ reduction using 1,000 pulses at 20 kV/cm (58). These published results in addition to our own support the well-established notion that bacteria are more resilient to PEF than mammalian cells. Additionally, our results indicate that MRSA biofilms are more resistant to nsPEF than planktonic bacteria. This was expected as bacteria in a biofilm structure are well protected due to the surrounding extracellular polymeric matrix (EPS). EPS protects encased bacteria from the host immune response and prevents antimicrobials from effectively permeating into the biofilm structure (96). Previous work on another skin pathogen, *Cutibacterium acnes*, indicated that biofilm-grown cells and intact biofilm were more susceptible to inactivation by PEF than free-living planktonic cells (97). This result did not replicate with MRSA. This could be due to the bacterial species having different EPS components that are more or less conducive to electrical currents or could be due to a more robust staphylococcal genetic response to cell envelope damage. The *S. aureus* biofilm matrix is composed largely of polysaccharides and proteins, although the components vary over time and differ depending on the nature of the biofilm substrate (98–100). To our knowledge, its conductive capacity has never been assessed. Similarly, this organism's transcriptional responses to stresses, including heat shock, cold shock, starvation, DNA damage, and oxidative stress, have been documented (101, 102), but to our knowledge, the transcriptional response to PEF has not been documented in this or any other prokaryotic organism.

When planktonic MRSA is treated with both nsPEF and sub-lethal concentrations of clinically relevant antibiotics, we see more robust bacterial killing by the combinatorial treatment than by either treatment administered as monotherapies. In the case of daptomycin, the relative order of nsPEF and antibiotic application is unimportant, as prior nsPEF treatment sensitizes cells to antibiotics, and prior antibiotic treatment appears to sensitize cells to nsPEF. This is consistent with a model in which nsPEF and daptomycin are both primarily active at the cell membrane, such that their effects reinforce each other regardless of which stress the cell encounters first. Excitingly, combination with nsPEF gives 2 µg/mL of daptomycin the same efficacy as 8 µg/mL of daptomycin applied alone. Doxycycline was modestly enhanced by combination with nsPEF, although the efficacy of dual therapies depended on the order of administration. nsPEF treatment modestly sensitized MRSA to doxycycline but the antibiotic did not substantially sensitize the bacteria to subsequent nsPEF exposure. Similarly, pre-treatment with vancomycin did not sensitize MRSA to nsPEF but a prolonged exposure to the antibiotic after nsPEF greatly increased the efficacy of the combined treatment. Our results also show that nsPEF did not sensitize MRSA to methicillin, suggesting that nsPEF will not stimulate a response to an antibiotic to which the bacteria are already resistant. However, mechanisms of resistance involving membrane proteins such as efflux pumps may be impacted by permeabilization of the cell plasma membrane by nsPEF.

Our results suggest that nsPEF administered to surface-accessible SSTIs could lower the effective dose of antibiotics needed to treat an infection, allowing more effective treatment without increasing the dose of antibiotic administered and risk of amplified

side effects. By reducing the dose of antibiotic necessary to be effective, co-treatment with nsPEF could amplify the effects of standard antibiotic dosing to treat *S. aureus* infections, reducing the risk that tolerant persister cells could survive treatment and cause recurrent infection.

Future research will focus on testing the efficacy of nsPEF *in vivo* in a mouse model of SSTI. A range of PEF amplitudes and durations will be tested to minimize damage to healthy tissue and muscle contraction. If these results are replicated in animal and human studies, nsPEF co-treatment could allow the treatment of suitable infections with lower antibiotic doses, reducing drug side effects. SSTIs are accessible to physical intervention and are good candidates for nsPEF co-treatment, which could be adopted as a step-in wound and abscess debridement. Notably, the CellFX pulse generator used in our experiments has already received clearance from the FDA, Europe, and Canada for the treatment of benign skin lesions.

Much more work will be needed before such possibilities can be realized, but here we report that nsPEF in combination with antibiotic treatments not only increases bacterial inactivation but also reduces the concentration of antibiotics necessary for disinfection. Additionally, this is the first time that it has been shown that the order of antibiotic/PEF administration is important in bacterial inactivation, which will influence future treatment design.

## ACKNOWLEDGMENTS

This research was supported by a grant from Pulse Biosciences, Inc. to C.M. and E.B.P.

The authors would like to thank Dr. Olga Pakhomova and Dr. P. Thomas Vernier (Frank Reidy Research Center for Bioelectrics, Old Dominion University, Norfolk, VA, USA) for the valuable discussion and help provided during the study.

## AUTHOR AFFILIATIONS

[1]Frank Reidy Research Center for Bioelectrics, Old Dominion University, Norfolk, Virginia, USA

[2]Biomedical Sciences Program, Old Dominion University, Norfolk, Virginia, USA

[3]Department of Chemistry and Biochemistry, Old Dominion University, Norfolk, Virginia, USA

[4]Department of Electrical and Computer Engineering, Old Dominion University, Norfolk, Virginia, USA

## AUTHOR ORCIDs

Erin B. Purcell http://orcid.org/0000-0002-8736-0433
Claudia Muratori http://orcid.org/0000-0002-3359-164X

## FUNDING

| Funder | Grant(s) | Author(s) |
| --- | --- | --- |
| Pulse Biosciences, Inc. | | Erin B. Purcell |
| | | Claudia Muratori |

## AUTHOR CONTRIBUTIONS

Alexandra E. Chittams-Miles, Data curation, Formal analysis, Methodology, Writing – original draft, Writing – review and editing | Areej Malik, Data curation, Formal analysis, Methodology, Writing – original draft, Writing – review and editing | Erin B. Purcell, Conceptualization, Funding acquisition, Supervision, Writing – original draft, Writing – review and editing | Claudia Muratori, Conceptualization, Data curation, Funding acquisition, Supervision, Writing – original draft, Writing – review and editing

## ADDITIONAL FILES

The following material is available online.

### Supplemental Material

**Supplemental figures (Spectrum02992-23-s0001.pdf).** Fig. S1 to S3.

### Open Peer Review

**PEER REVIEW HISTORY (review-history.pdf).** An accounting of the reviewer comments and feedback.

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
