## [Reviewer comments · Microbiology Spectrum]

Microbiology Spectrum

Nanosecond pulsed electric fields increase antibiotic susceptibility in methicillin-resistant *Staphylococcus aureus*.

Alexandra Chittams-Miles, Areej Malik, Erin Purcell, and Claudia Muratori

Corresponding Author(s): Claudia Muratori, Old Dominion University

Review Timeline:

Submission Date:	August 14, 2023
Editorial Decision:	October 12, 2023
Revision Received:	October 25, 2023
Accepted:	October 31, 2023

Editor: Brian Conlon

Reviewer(s): The reviewers have opted to remain anonymous.

Transaction Report:

DOI: <https://doi.org/10.1128/spectrum.02992-23>

October 12, 2023

Dr. Claudia Muratori
Old Dominion University
4211 Monarch Way
Norfolk

Re: Spectrum02992-23 (Nanosecond pulsed electric fields increase antibiotic susceptibility in methicillin-resistant *Staphylococcus aureus*.)

Dear Dr. Claudia Muratori:

Thank you for submitting your manuscript to Microbiology Spectrum. As you will see your paper is very close to acceptance. Please modify the manuscript along the lines I have recommended. As these revisions are quite minor, I expect that you should be able to turn in the revised paper in less than 30 days, if not sooner. If your manuscript was reviewed, you will find the reviewers' comments below.

When submitting the revised version of your paper, please provide (1) point-by-point responses to the issues raised by the reviewers as file type "Response to Reviewers," not in your cover letter, and (2) a PDF file that indicates the changes from the original submission (by highlighting or underlining the changes) as file type "Marked Up Manuscript - For Review Only". Please use this link to submit your revised manuscript. Detailed instructions on submitting your revised paper are below.

Link Not Available

Sincerely,

Brian Conlon

Reviewer comments:

Reviewer #1 (Comments for the Author):

The manuscript describes the application of electrical pulses of sub-microsecond pulse duration (nsPEF) to treat antibiotic resistant bacteria. Bacteria inactivation is an important problem, and electrical pulses were suggested for such treatment long time ago. While technical part of the manuscript is presented in sufficient details, there is a clear need to improve the presentation to make it more readable.

1. Introduction needs to be expanded. The problem is important, but authors need to show where their proposed method stands with respect to other approaches (see, for example, <https://doi.org/10.1073/pnas.2208378119>), and what is particularly new in this manuscript with respect to a number of prior publications in the field.
2. Experimental setup needs to be presented in the main text together with the workflow protocol to help readers' comprehension.
3. The structure of the manuscript and logical sequence is not clear and needs to be moved upfront to guide readers through manuscript.
4. Figures are difficult to comprehend based on rather brief captions which do not necessarily explain what is plotted there (axes, titles etc.). Authors need to revisit those figures and make sure that all of those together with their captions are self-explanatory.
5. Using log numbers in the text is confusing. log₁₀ for example ... Isn't it 1?
6. The choice of parameters for antibiotic concentration and number of pulses is unclear. More justification is needed.
7. Through out the text 30 kV/cm is mentioned but no indication is provided what was the actual electric field delivered to those

cells. No attempt of changing this electric field is reported or discussed.

8. Temperature rise is reported; however, the overall discussion of the effect of temperature is vague and is not logically sound.

9. Practical implementation of this approach for treating bacteria is unclear. It is suggested that authors discuss how they see this technique being implemented in clinical setting.

Reviewer #2 (Comments for the Author):

The manuscript was very logical and well thought out. The datasets were clearly presented and the data was not over described. The overall results are in line with nsPEF literature on the impact of bacterial treatment with nsPEF and antibiotics that has been performed by other groups. This specific paper focused on antibiotic resistant strains, which is much more clinically relevant as opposed to just an alternative to antibiotics. The chosen bacteria studied is also usually located on the surface of the skin so nsPEF has relevance as a treatment as opposed to much deeper wide spread infections. The authors were very intentional about the tests performed across multiple antibiotics and showed promising results in some applications and rather limited value in others. I applied the authors for showing not only what worked well, but also where nsPEF offered no benefit. The materials and methods, figures, and discussion was also informative and concise sticking many to the results of the paper and not extrapolating widely about the applications of this concept in clinical practice.

Preparing Revision Guidelines

Please return the manuscript within 60 days; if you cannot complete the modification within this time period, please contact me. If you do not wish to modify the manuscript and prefer to submit it to another journal, please notify me of your decision immediately so that the manuscript may be formally withdrawn from consideration by Microbiology Spectrum.

Point-by-point response to Reviewers

We thank the reviewers for their interest in our work and useful comments which helped us to improve the quality of our manuscript.

Response to Reviewer #1

1. Introduction needs to be expanded. The problem is important, but authors need to show where their proposed method stands with respect to other approaches (see for example <https://doi.org/10.1073/pnas.2208378119>, and what is particularly new in this manuscript with respect to a number of prior publications in the field.

We thank the reviewer for raising this point and are happy to provide more information. We have enhanced the Introduction by expanding the paragraph about other physical methods, explaining in more detail the difference between nanosecond pulses and conventional micro-and millisecond pulses, and clarified the novel aspects of our research; see page 5, line 74 and page 6, line 101.

2. Experimental setup needs to be presented in the main text together with the workflow protocol to help readers' comprehension.

We agree with the reviewer and decide to add a new figure (current supplementary figure 2) which includes a diagram of the experimental workflow, the nsPEF exposure system, and the 600 ns waveform. Please notice that while revising the manuscript we realized that the electric field was miscalculated, cell samples were exposed to 15 kV/cm not 30 kV/cm 600 ns pulses. See Supplementary Fig.2

3. The structure of the manuscript and logical sequence is not clear and needs to be moved upfront to guide readers through manuscript.

While reviewer 2 had the opposite impression and wrote “The manuscript was very logical and well thought out,” we realize that this might have been because they had more personal familiarity with the topic. We thank reviewer 1 for pointing out areas that needed to be amended to make the manuscript more accessible for a general audience. We have rephrased some hypotheses for clarity and further emphasized the connections between our findings in the Results section.

4. Figures are difficult to comprehend based on rather brief captions which do not necessarily explain what is plotted there (axes, titles etc.). Authors need to revisit those figures and make sure that all of those together with their captions are self-explanatory.

We have amended our figure legends for greater accessibility. Legends for Figure 1, Supplementary figure 1 and 3 have been revised to provide more details, see page 26, line 714 and page 31, line 824.

5. Using log numbers in the text is confusing. log₁₀ for example ... Isn't it 1?

We politely disagree with the reviewer. Inactivation rates are expressed as log₁₀ (CFU/mL_{control} - CFU/mL_{treated}). Log₁₀ CFU/mL is a standard unit of measure in

microbiology literature, used to express the relative number of living bacteria eliminated by an antimicrobial treatment. For recent examples please see (1) (2). We have edited our text to replace log10 with log₁₀, and hope this will make the distinction between log₁₀ units and the measurements themselves more clear.

6. The choice of parameters for antibiotic concentration and number of pulses is unclear. More justification is needed.

Antibiotic concentrations were based on both results from growth curves over 16 hours (Supplementary Figure 1) and CFU/mL after transient incubations with a wide range of antibiotic concentrations (Figure 3A, 5A, 6A).

Similarly a wide range of pulse numbers was initially tested (see figure 2). We decided to use 60 and 120 pulses (600 ns) because these conditions significantly perturbed MRSA without heating the cell sample beyond optimal *S. aureus* growth conditions, see page 8, line 145 and table 1.

The text has been clarified to address these points.

7. Through out the text 30 kV/cm is mentioned but no indication is provided what was the actual electric field delivered to those cells. No attempt of changing this electric field is reported or discussed.

We thank the reviewer for raising this point, which was addressed in our response to point 2. Unfortunately with the current setup we are not able to test higher fields in cuvettes. We recognize that this is an important point, and we are planning on testing more pulse amplitudes *in vivo* in animals as discussed in the discussion, see page 20, line 416.

8. Temperature rise is reported; however, the overall discussion of the effect of temperature is vague and is not logically sound.

We clarified the text, see page 8, line 145. We also modified Table 1 to report temperature changes for all 600 ns pulse doses (0, 60, 120 and 250 pulses).

9. Practical implementation of this approach for treating bacteria is unclear. It is suggested that authors discuss how they see this technique being implemented in clinical setting.

Discussion have been implemented to address this point, see 19, line 410.

Response to Reviewer #2

The manuscript was very logical and well thought out. The datasets were clearly presented and the data was not over described. The overall results are in line with nsPEF literature on the impact of bacterial treatment with nsPEF and antibiotics that has been performed by other groups. This specific paper focused on antibiotic resistant strains, which is much more clinically relevant as opposed to just an alternative to antibiotics. The chosen bacteria studied is also usually located on the surface of the skin so nsPEF has relevance as a treatment as opposed to much deeper wide spread

infections. The authors were very intentional about the tests performed across multiple antibiotics and showed promising results in some applications and rather limited value in others. I applied the authors for showing not only what worked well, but also where nsPEF offered no benefit. The materials and methods, figures, and discussion was also informative and concise sticking many to the results of the paper and not extrapolating widely about the applications of this concept in clinical practice.

We want to thank you the reviewer for her/his appreciation of our work.

1. Tello-Diaz, C., M. Palau, E. Munoz, X. Gomis, J. Gavalda, N. Fernandez-Hidalgo, and S. Bellmunt-Montoya. 2023. Methicillin-Susceptible *Staphylococcus aureus* Biofilm Formation on Vascular Grafts: an In Vitro Study. *Microbiol Spectr* 11: e0393122.
2. Bhavnani, S. M., J. P. Hammel, E. A. Lakota, M. Trang, J. C. Bader, C. C. Bulik, B. D. VanScoy, C. M. Rubino, M. D. Huband, L. Friedrich, J. N. Steenbergen, and P. G. Ambrose. 2023. Pharmacokinetic-Pharmacodynamic Target Attainment Analyses Evaluating Omadacycline Dosing Regimens for the Treatment of Patients with Community-Acquired Bacterial Pneumonia Arising from *Streptococcus pneumoniae* and *Haemophilus influenzae*. *Antimicrob Agents Chemother* 67: e0221321.

Re: Spectrum02992-23R1 (Nanosecond pulsed electric fields increase antibiotic susceptibility in methicillin-resistant *Staphylococcus aureus*.)

Dear Dr. Claudia Muratori:

Your manuscript has been accepted, and I am forwarding it to the ASM production staff for publication. Your paper will first be checked to make sure all elements meet the technical requirements. ASM staff will contact you if anything needs to be revised before copyediting and production can begin. Otherwise, you will be notified when your proofs are ready to be viewed.

Sincerely,
Brian Conlon
Editor
Microbiology Spectrum